# Transcriptional Response to Hypoxia: The Role of HIF-1-Associated Co-Regulators

**DOI:** 10.3390/cells12050798

**Published:** 2023-03-03

**Authors:** Angelos Yfantis, Ilias Mylonis, Georgia Chachami, Marios Nikolaidis, Grigorios D. Amoutzias, Efrosyni Paraskeva, George Simos

**Affiliations:** 1Laboratory of Biochemistry, Faculty of Medicine, University of Thessaly, BIOPOLIS, 41500 Larissa, Greece; angyfant@uth.gr (A.Y.); mylonis@med.uth.gr (I.M.); ghah@uth.gr (G.C.); 2Bioinformatics Laboratory, Department of Biochemistry and Biotechnology, University of Thessaly, BIOPOLIS, 41500 Larissa, Greece; marionik23@gmail.com (M.N.); amoutzias@uth.gr (G.D.A.); 3Laboratory of Physiology, Faculty of Medicine, University of Thessaly, BIOPOLIS, 41500 Larissa, Greece; fparaskeva@uth.gr; 4Gerald Bronfman Department of Oncology, Faculty of Medicine, McGill University, Montreal, QC H4A 3T2, Canada

**Keywords:** hypoxia, HIF-1, transcriptional regulation, chromatin, cancer

## Abstract

The Hypoxia Inducible Factor 1 (HIF-1) plays a major role in the cellular response to hypoxia by regulating the expression of many genes involved in adaptive processes that allow cell survival under low oxygen conditions. Adaptation to the hypoxic tumor micro-environment is also critical for cancer cell proliferation and therefore HIF-1 is also considered a valid therapeutical target. Despite the huge progress in understanding regulation of HIF-1 expression and activity by oxygen levels or oncogenic pathways, the way HIF-1 interacts with chromatin and the transcriptional machinery in order to activate its target genes is still a matter of intense investigation. Recent studies have identified several different HIF-1- and chromatin-associated co-regulators that play important roles in the general transcriptional activity of HIF-1, independent of its expression levels, as well as in the selection of binding sites, promoters and target genes, which, however, often depends on cellular context. We review here these co-regulators and examine their effect on the expression of a compilation of well-characterized HIF-1 direct target genes in order to assess the range of their involvement in the transcriptional response to hypoxia. Delineating the mode and the significance of the interaction between HIF-1 and its associated co-regulators may offer new attractive and specific targets for anticancer therapy.

## 1. Introduction: The Cellular Response to Hypoxia and the Role of HIFs

Hypoxia or lack of sufficient oxygen can occur under either physiological or pathological conditions such as intense muscular exercise or ischemic diseases, respectively. Hypoxia also characterizes the micro-environment of solid tumors and potentiates the aggressiveness and resistance of cancer cells to therapy. A key element in the cellular response to hypoxia is the stabilization of the alpha subunits of the hypoxia inducible factors (HIFα) and the subsequent activation of the HIF heterodimers, that upregulate the transcription of many genes required for adaptation at low oxygen conditions. The HIF family of heterodimeric transcription factors comprises three HIFα members (HIF-1α, HIF-2α, and HIF-3α) and one HIFβ member (HIF-1β, also known as aryl hydrocarbon receptor nuclear translocator, ARNT). HIF-1 is the most widely expressed and best studied form and it will be the subject of this review.

The breakthrough work by G. Semenza, Sir P. Ratcliffe and W. Kaelin (2019 Nobel prize in Physiology or Medicine) led to the characterization of the cellular oxygen sensing mechanism that controls the expression levels of HIFα [1,2,3,4,5]. Briefly, under atmospheric oxygen concentrations (normoxia), oxygen sensitive enzymes hydroxylate HIFα and cause its degradation and/or block its binding to transcriptional co-activators. The inactivation of these enzymes under hypoxia leads to stabilization of HIFα, its translocation into the nucleus, the formation of functional HIF heterodimer with ARNT, through their Per-Arnt-Sim (PAS) homology domains, and binding to specific DNA sequences called hypoxia response elements (HRE), through their basic helix-loop-helix (bHLH) domains. Thus, the transactivation domains (TAD) of HIFα can then interact with transcriptional coactivator proteins such as CREB-binding protein (CBP) and stimulate expression of genes containing HREs in the promoter or enhancer regions. The way HIF-1 selects the HREs it binds to, the means of its interaction with chromatin and chromatin-associated regulators and how these interactions may be controlled by oxygen-dependent or independent mechanisms are questions addressed in the following sections.

## 2. The HIF-Dependent Transcriptional Response

Early analysis of several different individual validated hypoxia-responsive and HIF-dependent target genes, revealed that the HRE comprises the short core consensus sequence 5′-RCGTG-3′, as originally determined in the erythropoietin enhancer, which led to the first purification and identification of HIF-1 [6,7]. In addition, early transcriptomic analyses using microarrays in different cell lines identified 500–4000 genes that changed their expression after exposure to hypoxia, while studies using chromatin immunoprecipitation (ChIP) coupled with analysis on microarrays (ChIP-chip) identified a much smaller number (approx. 300–500) of HIF-1 binding sites [8,9,10,11,12,13]. Several important conclusions were drawn from these studies.

First, a surprisingly small overlap between genes deregulated by hypoxia was detected among different cell types, suggesting that the transcriptional response to hypoxia depends a lot on cellular context [9,11,13]. Second, the majority of hypoxia responsive genes did not contain a detectable HIF-binding site in their proximal promoter, although the majority of HIF-1-binding sites were localized in close proximity to genes [9,10,12]. This indicates that a significant part of the transcriptional response to hypoxia is only indirectly regulated by HIF-1 through induction of other transcriptional regulators, in agreement with the observed large difference between the number of deregulated genes and the number of true HIF-1 binding sites. Furthermore, HIF-1-binding sites were mostly absent from genes down-regulated by hypoxia, suggesting that HIF-1 functions predominantly or even solely as a transcriptional activator [10,12]. Therefore, any transcriptional repression observed under hypoxia must be a result of HIF-1-dependent induction of repressor proteins and/or non-coding RNAs. Third, less than 1% of the DNA promoter sequences containing the core RCGTG motif bound HIF-1 or HIF-2 [10] and extended sequence preferences beyond the core motif could not explain the lower than-predicted number of observed HIF-1-bound sites [12], raising the issue of how productive HREs are selected. In relation to this, although many loci containing the core motif bound both HIF-isoforms, substantially more bound HIF-1 than HIF-2 [10]. This was in agreement with the considerably smaller contribution of HIF-2 to the transcriptional responses to acute hypoxia [8], at least under the conditions and cell lines studied, further underlining the question of selectivity.

Subsequent and more detailed studies utilizing RNA-seq and/or Chip-Seq [14] in combination with analysis of the non-coding transcriptome [15] and the role of HIF-α hydroxylases [16] or HIF-α isoforms [17] in many different cell lines [18,19] largely corroborated and extended the previous conclusions. These studies confirmed that only a relatively small set of genes (less than 50) are upregulated consistently and substantially by hypoxia or hydroxylase inhibitors in different human cell types, which may form the core of a hypoxia responsive gene signature [16,18,19]. It was also shown that, at genome-wide level, HIF-binding sites were enriched in the vicinity of gene promoters and their majority overlapped with DNAse1-hypersensitive peaks, i.e., open chromatin, although only approx. 1% of hypersensitive RCGTG motifs were bound by HIFs, indicating again that functional HREs may be defined by epigenetic mechanisms [14]. Interestingly, despite the fact that HIF-1 and HIF-2 share a common consensus DNA-binding motif, they were shown to bind different but overlapping sets of sites in chromatin and transactivate only partially overlapping sets of genes, in accordance with their distinct physiological functions and roles in disease [17]. HIF-1 binding sites were more often close to transcription start sites than those of HIF-2 and the binding site distribution was suggested to be caused by inherent properties of each isoform rather than by the severity or the duration of the hypoxic stimulus itself [17]. Concomitant analysis of RNA Pol II binding and histone H3 modification suggested that both HIFs may act predominantly through release of pre-bound promoter-paused RNA Pol II [15]. However, HIF-1 associated more strongly with histone H3 modifications (H3K4me3 and H3K9ac) that mark primarily promoters and proximal regulatory elements while HIF-2 interacted more strongly with H3 modifications (H3K4me1 and H3K27ac) often found in enhancers and other distal regulatory elements [17]. These studies suggested that functional HREs may be largely defined by preformed chromatin structures (i.e., present also under normoxia) which are not affected by HIF binding.

Overall, the genome-wide transcriptomic studies support the idea that HIFs do not alter the chromatin accessibility by their binding but rather associate with already defined and partially active promoters or enhancers, as also suggested by the fact that most HIF-target genes display normoxic expression which is further enhanced by hypoxia [20,21]. However, this is not an absolute rule as recent studies utilizing other than ChIP-seq methodology such as Micrococcal Nuclease (MNase) protection assays [22] and Assay for Transposase-Accessible Chromatin (ATAC)-seq [21] suggest that HIF binding at certain genes can also have a significant effect on nucleosome organization and chromatin accessibility. In either case, isoform specificity, gene selection and cell-type differences cannot be explained by a simple HIF-HRE association and must be conferred through interactions between HIFs and distinct transcriptional and chromatin-associated cofactors. Indeed, recent single-gene studies have identified a significant number of HIF-1α physical partners, several of which are involved in transactivation and act as HIF-co-regulators [23].

## 3. HIF-1-Interacting Co-Regulators

A compilation of proteins identified in physical association with HIF-1α and affecting the activity of HIF-1, by modulating its transactivation ability and not the expression levels of HIF-1α, is shown in Table 1, together with any known effectors, and schematically in Figure 1. Table 1 also includes the HIF-1α protein domains, regions or amino acid residues involved in the association with co-regulators or effectors (whenever this information is available, see also Figure 1) as well as the cell lines or types in which these associations were detected, so this information will not be repeated in the following sections. Although this review is focused on HIF-1-interacting co-activators, HIF-2 will also be mentioned in cases of common interactors. The list of co-regulators includes acetyl-transferases, such as p300/CBP and Tat-interactive protein (TIP60), enzymes introducing or removing methylation, other epigenetic enzymes or readers, basic components of the transcriptional machinery, chromatin remodeling factors and other proteins with miscellaneous functions which will be briefly discussed in the following sections. Bioinformatic analysis of a comparison between the genes affected by these co-activators (in cases with available transcriptomic data) and a compilation of validated direct HIF-1 gene targets is also presented in the last section of this review. The involvement of other hypoxia-activated transcription factors that interact with HIF-1 to mediate context-specific gene activation will not be examined here as it has been previously reviewed [24].
cells-12-00798-t001_Table 1Table 1A list of HIF-1α-interacting HIF-1 co-regulators with their effectors.HIFα IsoformHIF-1α  Domain (Residues) InvolvedCo-RegulatorsEffectorsRef.Cell TypesAcetyl transferasesHIF-1αN-TAD (532–585)  C-TAD (776–826)p300/CBP (+)
[25,26,27,28]HEK293, HeLa,  Hep3B, HCT166HIF-1α/2αAsn-803/847FIH-1 (−)[29,30,31]HEK293, Hep3BHIF-1αLys-674SIRT1 (−) PCAF (+)[32]HEK293, Hep3B, HT1080HIF-1αNDSRC-1 (+)  Ref-1 (+)[33]HEK293, COS7HIF-1αND^#^MUC1 (+)[34]Pancreatic cancerHIF-1αC-TAD^#^PKA (+)[35]HeLa, cardiomyocytesHIF-1αN-term. (1–400)FABP5 (+)[36]HEK293, HepG2HIF-1αC-TAD^#^CITED2 (−)[37,38]Hep3BHIF-1αODD/N-TAD  (429–608)FHL2 (−)[39]HEK293, Hep3BHIF-1α/2αC-TADFGFR2 (−)[40]DU145, PC3HIF-1αC-TADEAF2 (−)[41]HEK293, ccRCCHIF-1α
Reptin (−)^#^G9a (+)[42]MCF7HIF-1αNDPontin (+)^#^G9a (+)  GLP (+)[43]MCF7HIF-1α
TIP60 (+)
[44]HCT116Methylation/demethylation enzymesHIF-1αC-term. (575–826)^#^ JMJD2C (+)
[45]HeLa, MDA-MB-435HIF-1αbHLH (17–70)^#^ JMJD1A (+)
[46,47]HUVEC, UBCHIF-1α/2αPAS-B (175–305)^#^TET1 (+)
[48,49]HEK293, H1299, FADUHIF-1αNDSET9 (+)
[50]HEK293, Hep3B, U2OSHIF-1α/2αN-term. (1–396)SET1B (+)
[51]HeLa, A549Other epigenetic enzymes & Epigenetic readersHIF-1α/2αND^#^ PADI4 (+)
[52]Breast cancer, HepatomaHIF-1α/2αND^#^ ZMYND8 (+)p300 (+)[53]Breast cancerTranscriptional machineryHIF-1αC-TADCDK8-  Mediator (+)AFF4 (+)  CDK9 (+)[54]HCT116HIF-1α/2αC-term. (531–826)TRIM28 (+)CDK9 (+)[55]Breast cancerChromatin remodeling factorsHIF-1α/2αNDBRG1 (+)
[56]HEK293, Hep3BHIF-1α/2αC-term. (531–826)CHD4 (+)p300 (+)[57]Breast cancerHIF-1αETD (616–658)  Ser641/643^#^ NPM1 (+)ERK1/2 (+)[58]HeLa, Huh7Other proteinsHIF-1α/2αC-term. (531–826)^#^PKM2 (+)PHD3 (+)[59]HeLa, Hep3B, RCC4JMJD5 (+)[60]HeLa, MCF7Digoxin (−)[61]Macrophages
[62,63]MacrophagesHIF-1α/2αC-term./ID (604–726)FBP1 (−)
[64]HEK293, ccRCC, HK-2, A549HIF-1α/2αNDPARP1 (+)
[65]K562, MLFHIF-1αN-term. (1–390)Filamin A (+)
[66]Melanoma, HeLa NIH 3T3, COS1, HEK293, U2OS(−)/(+) denote negative/positive co-regulator or effector; ^#^ denotes HIF-1 direct target; ND: Not Determined.

### 3.1. CBP/p300

Very early studies [25] implicated one of the most important coactivator proteins, p300/CBP [67], in the regulation of HIF transcriptional activity (Table 1). Highly homologous E1A-binding protein p300 and CREB-binding protein CREBBP or CBP (often referred as a single p300/CBP moiety) regulate chromatin structure through histone and other protein acetylation in the transcriptional machinery. It has been shown that p300/CBP forms a protein complex with HIF-1α that is induced under low oxygen availability, it is recruited to chromatin via binding to HIF-1α and acts as adaptor protein in order to induce transcription of hypoxia-responsive genes [25]. p300/CBP can also function as a protein scaffold that binds simultaneously different transcription factors and thus receive multiple input information and signals. Specifically under hypoxia, CBP, which co-localises with HIF-1α in intranuclear foci, was shown to mediate formation of HIF-1α complexes containing Steroid Receptor Coactivator-1 (SRC-1), another coactivator (see also below) [28].

HIF-1α has two distinct transactivation domains, termed N-terminal and C-terminal TADs (N-TAD and C-TAD respectively; Figure 1). HIF-1 C-TAD (amino acids 786–826) interacts with the CH1 domain of CBP/p300 in a hypoxia-dependent manner [26]. HIF-1 N-TAD (amino acids 531–575) also associates with endogenous CBP/p300 through its CH3 domain and although this interaction is essential for transactivation, it is weaker compared to the C-TAD/CH1 interaction [26]. Post-translational modifications of HIF-1 C-TAD, including S-nitrosation of HIF-1α Cys800 (Cys848 for HIF-2α) [68], phosphorylation of HIF-2α Thr844 [69] and hydroxylation of HIF-1α Asn803 [29,70] affect the ability of this domain to recruit CBP/p300. In general, CBP/p300 appears to play a vital role in the formation of a HIF “co-activator-some” by recruiting secondary molecular players in order to assist the HIF-dependent transcription initiation [27].

### 3.2. Protein Effectors Regulating the Interaction between HIF-1α and CBP/p300

As already mentioned, CBP/p300 is responsible for an orchestrated cooperation with a broad variety of proteins, which in turn facilitate the HIF-1α-CBP/p300 interaction, regulate the assembly of the transcriptional apparatus and, consequently, stimulate transcription initiation. The HIF-1α-CBP/p300 interaction and, therefore, HIF-1-dependent gene expression can be regulated by various different effectors, a representative compilation of which are also briefly presented below and shown in Table 1 and Figure 1.

#### 3.2.1. Post-Translational Modifications Affecting the HIF-1α-CBP/p300 Interaction

Instability and degradation of HIF-1α under normoxia is mediated by binding of the tumour suppressor protein von Hippel Lindau (pVHL) to HIF-1α. This binding is triggered via hydroxylation of two Pro residues in the oxygen dependent degradation (ODD) domain of HIF-1α by a family of HIF-α specific prolyl-hydroxylases or PHDs. Another hydroxylase, targeting Asn-803 in HIF-1α (Asn-847 in HIF-2α), was originally identified as protein interacting with HIF-1α and termed Factor Inhibiting HIF-1 (FIH-1) [30]. Modification of HIF-1α C-TAD by FIH-1 under normoxia abrogates the HIF-1α/p300 interaction and blocks the transactivation activity of HIF-1, even in the case that HIF-1α escapes pVHL-mediated ubiquitination and degradation [29,31]. Thus, FIH-1 together with the PHDs comprise the oxygen sensing system that regulates both stability and activity of HIF-α [71].

In addition to hydroxylation, the interaction of HIF-1α with p300 has also been suggested to be regulated by acetylation at Lys-674, which lies N-terminally and outside the C-TAD. Sirtuin 1 (SIRT1) has been shown to physically interact with HIF-1α and reverse the lysyl acetylation introduced by the p300/CBP-associated factor (PCAF) [32]. Interaction/deacetylation by SIRT1 represses HIF-1α activity by blocking p300 recruitment facilitated by PCAF. Interestingly, although SIRT1 also interacts with HIF-2α, it enhances rather than represses HIF-2 transcriptional activity. It has been suggested that SIRT1 may be part of a HIF-1-specific positive feedback loop in which stimulation of glycolysis by HIF-1 and cytoplasmic NAD^+^ reduction leads to transcriptional downregulation of SIRT1 and further activation of HIF-1 [32].

#### 3.2.2. Positive Protein Effectors of the HIF-1α-CBP/p300 Interaction

SRC-1 as well as transcriptional mediators/intermediary factor 2 (TIF2) are transcriptional co-activators of the p160 protein family. They can interact with various members of the nuclear hormone receptor family to promote activation of transcription [72] and associate with co-activators [73,74,75], in order to bridge receptor activation to the basal transcriptional apparatus and enhance transcription initiation. SRC-1 has also been shown to interact with HIF-1α in a hypoxia-dependent manner. Both SRC-1 and TIF2, can boost HIF-1α mediated transcriptional activity, acting synergistically with CBP [33]. Redox factor 1 (Ref-1), a dual-function protein harbouring both DNA repair endonuclease activity and cysteine reducing activity, potentiated the functional and physical interaction of HIF-1α with SRC-1 and CBP, suggesting that in hypoxic cells Ref-1 facilitates the recruitment of the CBP–SRC-1 coactivator complex by HIF-1.

Mucin 1 (MUC-1) is a transmembrane protein mainly expressed in epithelial and hematopoietic cells with aberrant expression in various types of cancer [76,77]. Its small cytoplasmic tail (MUC1-CT) can be released under certain stimulatory conditions and translocate into the nucleus where it can affect gene expression via its interaction with transcription factors. MUC1-CT was shown to physically associate with HIF-1α and enhance HIF-1 activity independent of its effect on HIF-1α expression levels [34]. Furthermore, MUC1 also interacted with p300, occupied promoters of hypoxia-target genes and, when overexpressed, stimulated hypoxia-dependent recruitment of both HIF-1α and p300 to glycolytic gene promoters. Interestingly, MUC1 is itself a target of HIF-1α [78], suggesting the operation of a positive feedback loop in cancer cells under hypoxia.

cAMP-dependent protein kinase A (PKA) was shown to interact with HIF-1α and phosphorylate Thr-63 and Ser-692, thereby inhibiting the proteasomal degradation of HIF-1α, independently of prolyl hydroxylation, and promoting HIF-1 target gene expression [35]. Furthermore, PKA could also stimulate binding of the C-TAD of HIF-1α to p300 by counteracting the inhibitory effect of Asn803 hydroxylation. However, the mechanism of this stimulation was not clarified. Interestingly, a catalytic subunit of PKA could be induced by hypoxia in a HIF-dependent manner in A540 lung carcinoma cells [79], which may indicate the operation of yet another positive feedback loop in certain cell types.

Finally, proteomic analysis of HIF-1-binding partners led to the identification of fatty acid-binding protein 5 (FABP5), the cytosolic transporter of oleic acid, as an interactor and positive effector of HIF-1α [36]. FABP5 is shown to upregulate HIF-1α mRNA translation while it can also associate physically with HIF-1α and activate HIF-1 transcriptional activity by inhibiting FIH-dependent hydroxylation and promoting p300 binding to the HIF-1α C-TAD. Therefore, induction of FABP5 by oleic acid can promote HIF-1 activity and reinforces its role in lipid biogenesis and storage under hypoxia [80].

#### 3.2.3. Negative Protein Effectors of the HIF-1α-CBP/p300 Interaction

CBP/p300-interacting transactivator 2 (p35srj/CITED2), a 30-kDa protein, was discovered as an interactor of the CH1 domain of CBP/p300. This interaction inhibited binding of HIF-1α C-TAD to the same site and blocked the transactivation potential of HIF-1 [37] p35srj/CITED2 can be transcriptionally induced by hypoxia in a HIF-1-dependent way suggesting that p35srj is part of a negative-feedback loop which can finetune the availability of p300 not only for HIF-1α, but also for other p300-CH1 interacting transcription factors.

FHL2, a member of the four-and-a-half LIM domain (FHL) protein family was shown to associate with HIF-1α and inhibit HIF-1 (but not HIF-2) transcriptional activity without affecting HIF-1α expression levels [39]. Interestingly, two more members of the same protein family, FHL1 and FHL3, inhibited the transcriptional activity of both HIF-1 and HIF-2, the former by binding to p300 and blocking the HIF-α/p300 interaction (mimicking the action of p35srj/CITED2) and the latter via an unidentified mechanism. The expression of all three FHL proteins was induced by hypoxia in a HIF-dependent manner, suggesting that they may be part of a negative feedback loop.

Fibroblast growth factor receptor 2 (FGFR2) was shown to physically associate with both HIF-1α and HIF-2α and to inhibit HIF-1 transcriptional activity without affecting protein expression levels of either HIF-1α and HIF-2α [40]. Furthermore, FGFR2 could bind to the HIF-1α C-TAD and cause dissociation of p300, thereby inhibiting recruitment of HIF-1α and p300 to a HIF-1 target promoter. As the interaction of FGFR2 with HIF-1α and HIF-2α was stronger under normoxia, it may act as a means of ensuring low transcriptional activity of HIFs in normal oxygen conditions.

EAF2 (ELL-associated factor 2) is a potential tumour suppressor that binds to and stabilizes pVHL, thereby supressing HIF-1 activity [81]. In a subsequent study by the same team, EAF2 was shown to associate with HIF-1α, but not HIF-2α, and suppress the transcriptional activity of HIF-1, but not that of HIF-2 [41]. This suppression was attributed to the disruption of the interaction between the C-TAD of HIF-1α and p300 independently of FIH-1 and Sirt1. Moreover, the same study revealed that expression of EAF2 is directly induced by HIF-1 in response to hypoxia, suggesting yet another negative feedback regulation loop.

### 3.3. The TIP60 Complex

Although the lysine acetyl-transferases p300/CBP are usually considered as the main HIF transcriptional coactivators, abrogation of the interaction between HIF-1α and p300/CBP was shown to affect the expression of only a subset of HIF target genes [82]. It appears that the human Tip60 or nucleosome acetyltransferase of histone H4 (NuA4) chromatin-remodelling complex, a multiprotein complex that consists of at least 16 subunits [83], also plays a significant role as HIF-1 co-activator. Two of the subunits of the Tip60 complex are Pontin and Reptin that belong to the family of AAA^+^ helicases (ATPases associated with diverse cellular activities) and participate in the control of transcription both as members of the Tip60 complex and independently through their association with a variety of transcription factors [84,85]. Both reptin and pontin as well as other Tip60 subunits act as HIF-1 co-regulators (Table 1 and Figure 1).

Reptin was initially reported to physically associate with HIF-1α (but not HIF-2α) and repress the expression of a significant number of HIF-1 target genes [42]. This activity required methylation of Reptin by lysine methyltransferase G9a, the expression of which is upregulated under hypoxia [86], suggesting the operation of a negative feedback loop. Methylated Reptin is recruited to HIF-1 target and Reptin-dependent promoters via its binding to HIF-1α and, in turn, recruits Histone Deacetylase 1 (HDAC1) and blocks association of RNA Pol II, thereby repressing transcription [42]. More recently, Reptin has been shown to also associate with HIF-2α, under ERK1/2 inactivating conditions that increase the cytosolic pool of HIF-2α. This association also results in lower HIF-2 activity, albeit, via a different mechanism that involves PHD/VHL-independent HIF-2α destabilization [87].

Like Reptin, methylation of Pontin by G9a and G9a-like protein (GLP) is stimulated under hypoxia and methylated Pontin physically interacts with HIF-1α [43]. However, unlike Reptin, methylation-dependent association of Pontin with HIF-1α resulted in transcriptional activation of a subset of hypoxia inducible genes. This involves interaction of Pontin with p300 and stimulation of its association with chromatin-bound HIF-1α. Pontin depletion affected a significant number of HIF-1 target genes, which did not overlap with the set of genes affected by Reptin. These opposing and Tip60-independent roles of Pontin and Reptin on different sets of HIF-1-target genes have been suggested to reflect the flexible ability of HIF-1α to interact with distinct co-activators/repressors under different cellular/environmental contexts.

Partly in contrast to the above, genetic experiments in Drosophila suggested that both Reptin and Pontin, as well as other subunits of the Tip60 complex are required for the transcriptional activity of Sima, the Drosophila homolog of HIF1A [44]. Experiments with human cancer cells confirmed that a significant proportion of HIF-1-dependent genes relied on Tip60 for their induction by hypoxia and demonstrated physical association between HIF-1α and components of the Tip60 complex, including Tip60 itself, the catalytic subunit with lysine acetyl transferase (KAT) activity, Reptin and Pontin. Recruitment of the Tip60 complex by HIF-1α promoted acetylation of histone H3 lysine 9 (H3K9) and histone H4 as well as activation of RNA Pol II by phosphorylation of its C-terminal domain (CTD) in Tip60-dependent promoters [44].

### 3.4. Methylation & Demethylation Enzymes

In addition to acetylation, histone and DNA methylation also play crucial roles in epigenetic regulation of gene expression and can severely affect the activity of transcription factors, including HIF-1 (Table 1 and Figure 1). Methylation of lysine residues on histone proteins is mediated by the histone lysine methyltransferases (KMTs) while the removal of methyl groups is accomplished by lysine demethylases (KDMs). An important KDM family are the Jumonji C (JmjC) domain-containing (JMJD) demethylases. A member of this family, JMJD2C, encoded by the *KDM4C* gene, was shown to interact via its catalytic domain with HIF-1α but not HIF-2α [45]. JMJD2C stimulated the transcriptional activity of HIF-1, but not HIF-2, and this effect depended on its histone demethylase activity. HIF-1 could mediate recruitment of JMJD2C specifically to HIF-1 target genes in hypoxic cells and JMJD2C could in turn enhance HIF-1 binding to the HREs of target genes, demethylate H3K9me3 at these sites and stimulate their expression. The *KDM4C* gene is a HIF-1 target gene [88]; therefore the interaction of JMJD2C with HIF-1α may provide a positive feedback mechanism in cancer cells by amplifying HIF-1–mediated transactivation.

In addition to JMJD2C, HIF-1 is directly involved in the hypoxic induction of another two members of the JMJD family of demethylases, JMJD1A (KDM3A) and JMJD2B [88,89]. Similarly to JMJD2C, JMJD1A was shown to associate with HIF-1α, be recruited by HIF-1α to the enhancer of SLC2A3, a direct target of HIF-1 encoding for glucose transporter 3 (GLUT3), and demethylate the repressive histone H3K9me2 [46]. Similar observations were also made for the promoter of PGK1, the gene encoding for the glycolytic enzyme phosphoglycerate kinase 1 [47]. Both JMJD1A studies suggested that the cooperation between HIF-1 and JMJD1A was especially important for the hypoxic induction of glycolytic genes and the subsequent upregulation of glycolysis in both normal and cancer cells.

Demethylation of DNA involves the ten-eleven-translocation (TET) family of 5-methylcytosine dioxygenases that catalyze the conversion of 5-methylcytosine (5-mC) to 5-hydroxymethylcytosine (5-hmC) and can regulate gene expression both dependent and independent of their catalytic activity [90]. Both HIF-1 and HIF-2 have been shown to mediate the hypoxic induction of TET1 in various cell lines [49]. Furthermore, TET1 as well as an enzymatically inactive TET1 mutant were shown to associate physically with both HIF-1α and HIF-2α and to enhance their transactivation activities. Knockdown of TET1 alleviated the hypoxia induced epithelial-mesenchymal transition, a phenotype rescued by the catalytically inactive TET1 mutant. Therefore, the HIF-co-activator function of TET1 was independent of 5-hmC formation, although TET1 could demethylate hypoxia-inducible promoters. Indeed, a concurrent study could demonstrate that TET1 expression was essential for the global 5-hmC gains and the subsequent upregulation of HIF-1 target genes observed under hypoxic conditions [48].

A major family of KMTs are the SET (Su(var)3-9, Enhancer of Zeste, Trithorax) domain containing histone methyltransferases. A member of this family, SET9, was shown to associate with HIF-1α, but not HIF-2α [50]. SET9 could stabilize HIF-1α in various cells lines by inhibiting its proteasomal degradation. In addition, SET9 was enriched and was required for HIF-1α binding, H3K4 monomethylation and transactivation at a subset of HIF-1 target gene promoters regulating mostly expression of glycolytic genes, suggesting a selective role of the HIF-1α/SET9 interaction in the upregulation of glycolysis under hypoxia [50]. Another example of differential HIF-1-target gene regulation is provided by a second histone H3K4 methyltransferase, SET1B [51]. Unlike SET9, SET1B could associate with both HIF-1α and HIF-2α and its expression was required for the upregulation of a subset of hypoxia-inducible genes, which included both HIF-1 and HIF-2 targets but encompassed preferentially genes involved in angiogenesis and less in glycolysis. SET1B was recruited to HIF target sites by the HIF complex and mediated H3K4me3 deposition across the gene bodies, which subsequently also increased H3K27 acetylation. Therefore, it appears that association of HIF-1α with different KMTs may dictate gene selectivity, which may also relate to the fact that certain HIF-1 gene targets, such as glycolytic ones, are expressed to a certain degree also under normoxia, while others, such as angiogenic ones, are significantly expressed only upon exposure to low oxygen.

### 3.5. Other Epigenetic Enzymes: PADI4

In addition to acetylation and methylation, histone citrullination can also affect regulation of gene expression by modulating chromatin binding of transcription factors and co-factors in conjunction with other epigenetic marks [91]. Citrullination is the hydrolysis of arginine (Arg) residues to citrulline (Cit), which is catalyzed by the small family of peptidylarginine deiminase (PADI or PAD) enzymes, and can affect histone-histone, histone-DNA or histone-protein interactions. A member of this family, PADI4, was shown to be upregulated by as well as associate with HIF-1α and/or HIF-2α [52]. Furthermore, in a positive feedback way, PADI4 could be recruited to HREs by HIFs, stabilize HIF occupancy of HREs and activate HIF target gene transcription (Table 1 and Figure 1). Recruitment of PADI4 to HIF target genes was required for hypoxia-induced citrullination of histones H3 and H4 as well as for deposition of marks of actively transcribed chromatin such as H3K4me3, H3K36me3, H3K4ac, and H4K5ac. There was a very high overlap between hypoxia-inducible genes regulated by HIFs and those that depended on PADI4 expression, suggesting that, at least in cancer cells, PADI4 is a global co-activator of HIFs.

### 3.6. Epigenetic Readers: ZMYND8 & BRD4

Modified histones are recognized by epigenetic readers containing protein domains that bind to acetylated or methylated histone residues. ZMYND8 (Zinc finger MYND-type containing 8), a core chromatin reader/effector with affinity for acetyl and methyl lysine residues of histones H3 and H4 [92], was shown to physically interact with HIF-1α and HIF-2α, while its hypoxic expression was also regulated by HIF-1 and HIF-2 [53] (Table 1 and Figure 1). Moreover, ZMYND8 was required for the hypoxic upregulation of the majority of HIF-controlled genes without affecting protein levels of HIF-1α and HIF-2α, constituting yet another positive feedback mechanism that amplifies HIF mediated transactivation and subsequent breast cancer progression and metastasis. ZMYND8 colocalizes with HIFs on HREs and its positive effect on their activity requires its acetylation by p300 and its association with Bromodomain-containing protein 4 (BRD4), another bromodomain acetyl lysine reader [93], which can in turn lead to release of paused RNA Pol II and transcriptional elongation of HIF target genes.

### 3.7. Components of the Transcriptional Machinery

#### 3.7.1. CDK8-Mediator

The HIF-1 co-regulators described so far include epigenetic writers, erasers, and readers that can modify chromatin around HREs and/or upon HIF-1 binding and make it more accessible to the basic components of the transcriptional machinery. However, this may not necessarily suffice for activation of RNA Pol II and transcription initiation and elongation. As already mentioned, HIFs predominantly bind to open chromatin regions and activate promoters with pre-bound, paused RNA Pol II. [15]. A functional and physical connection between transcription factors and the basal transcriptional machinery, including RNA Pol II, is provided by the Mediator, a large conserved multi-subunit and modular complex, that can stimulate phosphorylation of the Pol II CTD and trigger Pol II release from promoters and transition from transcription initiation to productive elongation [94]. The Mediator comprises three core modules while a fourth, the cyclin-dependent kinase 8 (CDK8) module transiently associates with the rest of the complex. Productive elongation is also controlled by the super elongation complex (SEC) comprising the RNA Pol II elongation factors, including the positive transcription elongation factor (P-TEFb) and its catalytic subunit cyclin-dependent kinase 9 (CDK9), which also targets the CTD of Pol II [95].

CDK8 as well as other components of the mediator CDK8-module were shown to interact with the C-TAD of HIF-1α and to be recruited by HIF-1 to HIF-1-target promoters [54] (Table 1 and Figure 1). CDK8 was required for the upregulation of the majority of the hypoxia-inducible genes, suggesting that the Mediator is an important, and probably global, HIF-1 co-activator. Mechanistically, although HIF-1α binding to chromatin was independent of CDK8, the interaction between HIF-1α and CDK8 was required to attract AFF4, the scaffold subunit of the SEC, and CDK9, thereby triggering release of paused RNA Pol II and robust induction of HIF-1-dependent genes.

#### 3.7.2. TRIM28/DNA-PK

Release of paused RNA Pol II at signal-regulated genes can also be controlled by the tripartite motif containing protein 28 (TRIM28), which stabilizes paused Pol II but also triggers Pol II release when phosphorylated on S824 by either DNA-dependent protein kinase (DNA-PK) or the kinase ataxia telangiectasia mutated (ATM) [96]. TRIM28 as well as the three subunits of DNA-PK were shown to associate with HIF-1α and HIF-2α [55] (Table 1 and Figure 1). Both TRIM28 and the catalytic subunit of DNA-PK (DNA-PKcs), which is activated by hypoxia through phosphorylation, were required for HIF-1 transcriptional activity, hypoxia-induced expression of known HIF-1-target genes, and stable HIF occupancy of their HREs. Furthermore, most of the genes induced in HIF-dependent manner were also TRIM- and DNA-PKcs-dependent and the corresponding mRNAs were enriched for mediators of glycolysis and angiogenesis. Recruitment of TRIM28 and DNA-PK by HIF-1 led to phosphorylation of TRIM28 by DNA-PK, which could subsequently promote interaction with CDK9, release of paused Pol II, and productive transcriptional elongation of HIF target genes in response to hypoxia.

### 3.8. Chromatin Remodeling Factors

#### 3.8.1. BRG1

In addition to DNA and histone modifications, chromatin topology, DNA-accessibility and, therefore, gene activation are also controlled by ATP-depended remodeling. The mammalian Switch/Sucrose-Nonfermentable (mSWI/SNF) chromatin-remodeling complexes contain the Brahma-related gene 1 (BRG1) or Brahma (BRM) proteins that have DNA-stimulated ATPase activity and can remodel chromatin through nucleosome sliding and eviction [97]. BRG1 was shown to associate with HIF-1α and HIF-2α and be recruited by HIF to a subset of HIF-target genes causing nucleosome remodeling and stimulation of transcription in hypoxic cells [56] (Table 1 and Figure 1). Interestingly, in the same study, evidence was presented suggesting that the BRG1 complex may also be involved in expression of the genes coding for HIF-1α and HIF-2α themselves.

#### 3.8.2. CHD4

Chromatin remodeling ATPase activities and histone deacetylation are coupled in the nucleosome remodeling and deacetylase (NuRD) complex, which is defined by the presence of one of the chromodomain helicase DNA-binding proteins 3–5 (CHD3–5) [98]. CHD4 was shown to form a complex with both HIF-α isoforms and could potentiate HIF-dependent transactivation in a subset of HIF target genes [57] (Table 1 and Figure 1). Interestingly, the effect of CHD4 on HIF-mediated transcription was independent of its helicase activity and other subunits of the NuRD complex. Nevertheless, HIFs mediated recruitment of CHD4 to HIF target genes and CHD4 itself also assisted HIF-1 occupancy on HREs. CHD4 could also associate with RNA Pol II through p300 under both normoxia and hypoxia, suggesting that CHD4 may facilitate loading of paused RNA Pol II on HIF targets under normoxia and its subsequent release upon HIF binding under hypoxia.

#### 3.8.3. NPM1

Nucleophosmin (NPM1) is a nuclear protein usually associated with nucleolar ribosomal biogenesis [99]. However, NPM1 is also involved in transcriptional activation, as NPM1 interacts both with chromatin components, through its histone chaperone ability, and transcription factors such as the nuclear factor kappa-light-chain-enhancer of activated B cells (NF-κB) [100]. NPM1 was recently identified as a HeLa cell protein binding, through its C-terminal aromatic domain, to a C-terminal HIF-1α region containing the ERK1/2 modification sites at Ser641/643 [101], termed ETD (ERK-target domain) and conserved in HIF-1α but not HIF-2α (Table 1 and Figure 1). This interaction was direct, as shown by in vitro binding of purified recombinant protein fragments, and stimulated inside cells by ERK1/2-mediated phosphorylation of HIF-1α or the phosphomimetic mutation Ser641Glu [58]. NPM1 apparently participates in a HIF-1 specific positive feedback mechanism as its hypoxic expression is mediated by HIF-1 [102] and its association with the ETD domain of HIF-1α enhances HIF-1 transcriptional activity and upregulates the hypoxic expression of several HIF-1, but not HIF-2, gene targets. NPM1 occupied HRE-containing and HIF-1-dependent promoters under normoxia and its presence was essential for subsequent recruitment and stable binding of HIF-1 under hypoxia. HIF-1α- and NPM1-regulated genes significantly overlapped and disruption of the HIF-1/NPM1 association with cell penetrating peptides derived from the ETD sequence of HIF-1α inhibited cancer cell proliferation and survival under hypoxia by triggering apoptosis [103]. The interaction between HIF-1 and NPM1 is the only known so far to be directly stimulated by an oncogenic pathway such as the MAPK/ERK that controls cellular proliferation. Oligomeric NPM1 is known to undergo liquid-liquid phase separation [99], so its association with HIF-1 target genes and HIF-1 itself (the C-terminal part of which is predicted to be largely intrinsically disordered) may contribute to the formation of a phase-separated multi-molecular assembly of transcriptional co-activators [104,105] that maintains hypoxia-inducible promoters in an open and rapidly activated state.

### 3.9. Other Proteins

#### 3.9.1. PKM2

Pyruvate kinase (PK) catalyzes the last and irreversible step in glycolysis, the conversion of phosphoenolpyruvate to pyruvate with the simultaneous production of ATP. The PKM gene encodes two distinct isoforms, PKM1 and PKM2, through alternative splicing. Expression of PKM2 is associated with rapidly proliferating cells and is thought to promote the Warburg effect of cancer cells and tumorigenesis [106]. PKM2, but not PKM1, as well as a catalytically inactive PKM2 mutant were shown to associate physically with HIF-1α and HIF-2α and to promote HIF-1 and HIF-2 transcriptional activity, without affecting their protein expression levels [59] (Table 1 and Figure 1). The HIF-1α/PKM2 interaction and PKM2-mediated HIF-1α transactivation required prolyl hydroxylation of PKM2 by PHD3. PKM2 and PHD3 both colocalized with HIF-1α and enhanced its binding at the HREs of HIF-1 target genes. PKM2 also interacted with p300 and promoted its recruitment and H3K9 acetylation in the same HIF-1 target genes. Both the genes encoding for PKM2 and PHD3 are directly regulated by HIF-1 [59,107] suggesting their participation in a positive feedback loop, which may be especially important in cancer cells expressing HIF-1α under well-oxygenated conditions, i.e., when PHD3 is catalytically active.

PKM2 was also shown to associate with JMJD5, a JMJD protein with lysine demethylation and hydroxylation activity, and their complex promoted HIF-1 transcriptional activity in breast cancer cells, while JMJD5 and PKM2 were co-recruited to HREs of two known HIF-1 target genes enhancing HIF-1α binding [60]. The interaction of PKM2 with HIF-1 and its positive effect on HIF-1-dependent transcription of glycolytic genes or the IL-1β gene were also demonstrated in LPS-stimulated macrophages [62,63] and binding of digoxin, a cardiac glycoside, to PKM2 resulted in chromatin remodeling and downregulation of HIF-1α transactivation, independently of PKM2 kinase activity, in the same type of cells [61], which may have therapeutic implications in inflammatory diseases.

#### 3.9.2. FBP1

Another metabolic enzyme involved in HIF regulation is FBP1 (fructose 1,6, bisphosphatase), that catalyzes the hydrolysis of fructose 1,6, bisphosphatase to fructose 6-phosphate in the penultimate irreversible and regulatory step of gluconeogenesis. FBP1 was shown to bind directly to the C-terminal regions of HIF-1α and HIF-2α [64] (Table 1 & Figure 1), with the interaction site on HIF-1α mapped using purified recombinant proteins in the, so called, inhibitory domain (ID; [108]) that lies between the N-TAD and the C-TAD and contains the nuclear localization signal [109], the nuclear export signal [110], the ERK1/2 phosphorylation sites [101] discussed above and many other modification and regulatory sites [111]. This interaction did not require the catalytic activity of FBP1, could take place on the HRE-containing promoters of HIF-1-target genes and repressed HIF activity, expression of HIF-target genes, glucose catabolism and hypoxic cellular proliferation, explaining the tumor suppressive functions of FBP1 [64].

#### 3.9.3. PARP1

Poly (ADP-ribose) polymerase 1 (PARP1) belongs to a large family of enzymes that can synthesize long and branched polymers of ADP-ribose onto acceptor proteins. PARP1 has DNA-binding and catalytic activities important for DNA repair, histone modification, chromatin remodeling and gene expression [112]. PARP1 was shown to associate with HIF-1α, bind in recombinant form directly to HIF-1α and HIF-2α, and promote HIF-dependent transcription, which required PARP1 catalytic activity but without affecting HIF-1 expression or its binding to DNA [65] (Table 1 and Figure 1).

#### 3.9.4. Filamin A

Filamins are large actin-binding proteins that stabilize the actin cytoskeleton and connect it to the plasma membrane. Filamin A (FLNA), the most abundant of the three filamin isoforms, interacts with a wide range of proteins involved in signal transduction, including transcription factors. It has been known that cleavage of FLNA by calpain, generates a C-terminal fragment that enters the nucleus and can modulate the activity of transcription factors [113]. The C-terminal fragment of FLNA, the release of which was enhanced under hypoxia, was also shown to interact with HIF-1α, but not HIF-2α [66] (Table 1 and Figure 1). Furthermore, FLNA promoted HIF-dependent transcriptional activity and its C-terminal fragment was recruited to the *VEGF-A* promoter, enhancing binding of HIF-1 and its transactivation function.

## 4. Defining a Core of HIF-1α-Associated HIF-1 Co-Activators

As mentioned above, in several cases of the aforementioned co-regulators, transcriptomic analysis demonstrated a significant overlap between the co-regulator-dependent genes and the HIF-1-dependent genes, the latter usually being defined as the hypoxia-inducible genes that are de-regulated upon depletion of HIF-1α. However, as also explained in Section 2 above, only a fraction of these genes are true direct HIF-1 targets, with the majority of them being only indirectly regulated by HIF-1 through HIF-1-dependent induction of other transcriptional regulators. Therefore, to get a better idea of the extent of co-regulator involvement in HIF-1-driven transcriptional regulation, we first screened the literature and defined a group of 83 highly validated HIF-1-dependent genes as List A (Appendix A). This list contains genes that were characterized as HIF-1 direct targets in single gene studies and satisfied the following four stringent criteria: a. they were inducible under hypoxia at both mRNA and protein levels; b. they were down-regulated by silencing HIF-1α under hypoxia; c. their promoters were activated by HIF-1 under hypoxia or HIF-1α stabilization conditions in gene reporter assays; d. HIF-1 binding to their promoters could be detected by ChIP assays or the functionality of the HREs present in their promoter were verified by mutational analysis.

In addition, we also re-analyzed publicly available transcriptomic data from eight genome-wide studies which used microarrays, ChIP to chip, ChIP-Seq, RNA-seq or bioinformatic analysis to define HIF-1-dependent genes [8,9,10,11,12,13,14,114] and compiled a list of 433 potential HIF-1 targets, 49 of which were also among the genes comprising List A. Of the rest, 109 genes that were identified in at least two of the aforementioned eight genome-wide studies (but not in any of the single-gene studies) were included in List B as likely, but not verified, HIF-1 direct targets (Appendix A). Next, List A and List B gene datasets were analyzed in comparison with differentially expressed genes found to be controlled by HIF-1 co-regulators in studies with available and processible transcriptomic data (Table 2 and Appendix A). Gene datasets were compared in pairs (co-activator-dependent genes vs. List A or List B) using DeepVenn [115] to identify common genes.

As shown in Table 2, datasets of genes regulated by five HIF-1 co-activators, namely ZMYND8, CDK8, TRIM28, NPM1 and JMJD1A, were found to have significant overlap with both List A and List B genes. As revealed by KEGG pathway analysis performed with ShinyGO [116] and SciPy [117], the common genes between each of these five HIF-1 co-activator-dependent gene sets and List A were enriched for HIF-1 signaling, cancer-associated pathways and carbohydrate metabolism (Figure 2; see also Appendix A, left panels, for single co-regulator analysis). On the other hand, co-activator-dependent genes overlapping with List B were involved in a wider spectrum of pathways including metabolism, p53 signaling, and autophagy (Figure 2; see also Appendix A, right panels, for single co-regulator analysis), suggesting that List B genes may not be part of a core hypoxia signature, but rather displaying context specific expression. The overlap between List A or List B genes and all five of the aforementioned co-activator-dependent gene sets are shown in the Venn diagrams of Figure 3A,B.

There is high degree of overlap between List A and co-activator-dependent genes, with the majority of the List A genes (58/83, 70%) also found in at least one of the co-activator-dependent gene sets and more than half of List A genes (42/83, 51%) being common with two or more of the co-activator-dependent genes (Figure 3A and Appendix A). Although only four genes of List A (namely *ANGPTL4*, *BHLHE40*, *BNIP3L* and *SLC2A1*) are co-dependent on all five co-activators, it is very likely that these five factors comprise the core of a HIF-1-dependent transcriptional activator complex involved in the regulation of many HIF-1 direct gene targets. KEGG pathway analysis of the List A genes not found in any of the co-activator gene sets (25/83, 30%), shows, in addition to hypoxic or cancer signaling, their involvement in other, apparently unrelated, pathways (Figure 3C), indicating limited expression that may explain their non-detection in the co-activator transcriptomic analyses.

The degree of overlap between List B and co-activator-dependent genes is similar but somewhat lower than List A, with the majority of the List B genes (80/109, 73%) also found in at least one of the co-activator-dependent gene sets but less than half of List A genes (48/109, 44%) being common with two or more of the co-activator-dependent genes (Figure 3B and Appendix A). List B genes not found in any of the co-activator gene sets (29/109, 27%) are associated with pathways not directly related to hypoxic signaling (Figure 3D), suggesting either limited expression or indirect regulation by HIF-1 and its associated co-activators

## 5. Concluding Remarks

The repertoire of transcription co-regulators that escort HIF-1α and consolidate its activity and functions is wide and still expanding. Besides revisiting the important role of HIF-1, the knowledge acquired from the studies reviewed in this article helps to draw a number of conclusions and better appreciate the complexity of transcriptional regulation by HIF-1. First, proteins with diverse functions, including histone modification enzymes, epigenetic readers, components of the transcriptional machinery as well as chromatin remodeling factors have been identified as HIF-1α interactors and co-regulators. By collectively looking at them (Table 1 and Figure 1), one can easily see that the vast majority of HIF-1α-interacting co-regulators enhance or are required for HIF-1-dependent transcriptional activation. This is probably to be expected, as cell survival under hypoxia depends on continuous transcription of HIF-1-target genes. If and when HIF-1 down-regulation is necessary, e.g., upon re-oxygenation or after chronic hypoxia exposure, it can easily be achieved by the oxygen-mediated activation or HIF-1-mediated transcriptional upregulation of PHDs, respectively, i.e., through the elegant system that regulates HIF-1α protein stability [5].

A second interesting point is that many of the HIF-1α co-activators are themselves direct targets of HIF-1 (Table 1 and Figure 1). This suggests the operation of several positive feedback loops that promote the activity of HIF-1, thus enhancing the robustness of the transcriptional response to hypoxia. On the other hand, a few cases of negative feedback regulation are mediated by HIF-1 direct targets that inhibit the interaction between HIF-1α and CBP/p300, probably in response to specific signaling pathways and in order to balance the co-activator function of CBP/p300 between different transcription factors. A third important observation is that the C-terminal part of HIF-1α harbors the binding sites for significantly more co-regulators than does the N-terminal part (Table 1 and Figure 1). This is not surprising as the conserved and highly structured N-terminal part of HIF-1α mediates both DNA binding (through the bHLH domain) and heterodimerization with ARNT (through the PAS domain), so it is hard to accommodate additional interactions without disturbing the vital associations with DNA and ARNT. Furthermore, the C-terminal part HIF-1α contains, in addition to the two TADs that by definition interact with co-regulators, the ID. This domain, even though it was originally described as TAD repressor [108], it apparently acts as a regulatory hub since it controls the subcellular localization of HIF-1α and harbors many modification sites [111] that facilitate fine-tuning of HIF-1 activity in response to signaling pathways, as exemplified by the ERK1/2 mediated control of the HIF-1α/NPM1 interaction [58]. Finally, the C-terminal part of HIF-1α, the overall three-dimensional structure of which remains unknown, is predicted to be largely intrinsically unstructured (https://alphafold.ebi.ac.uk/entry/Q16665 accessed on 31 January 2023), which could allow flexibility and multiple interactions through conformational adaptation elicited by residue modifications or physical contact with different partners.

A question arising in light of the large number of HIF-1α co-regulators, is whether and how these factors collaborate/interact with each other. So far, most co-regulators have been studied on their own for their effect on HIF-1, so, although this is not an easy task, it is important to study them in combination in order to, eventually, join the snapshots collected so far into a whole picture of HIF-1 at work on the target gene promoters. This is, probably, more achievable by analyzing the HIF-1α interactome on the promoters of single genes rather than trying to draw conclusions from genome-wide transcriptomic data that are often incomplete or cell-type specific. On the other hand, a question that also needs to be addressed is whether a HIF-1α co-regulator acts globally or is specific for a particular set of genes, which would require detailed genome-wide chromatin co-occupancy studies. Last but not least, it is urgent to map precisely and structurally characterize the physical interactions between HIF-1α and its co-regulators, in order to not only gain mechanistic insight but to also allow the design of drugs that can specifically target these interactions for therapeutic purposes.

## Figures and Tables

**Figure 1 cells-12-00798-f001:**
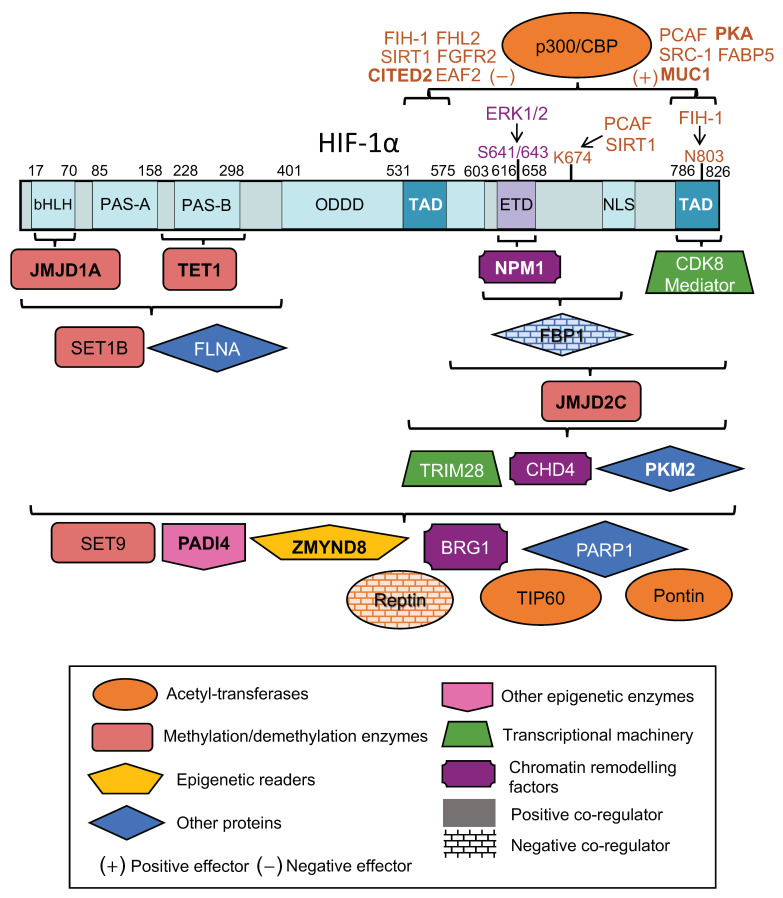
Schematic representation of HIF-1α and its interacting co-regulators. Positive (+) and negative (−) effectors of the p300/CBP-HIF-1α interaction are also shown. Brackets indicate the interacting region of HIF-1α, in cases that this has been experimentally defined. Residues, modification of which is known to affect an interaction, are also indicated. Genes directly regulated by HIF-1 are shown in bold. See Table 1 and text for details and relevant references.

**Figure 2 cells-12-00798-f002:**
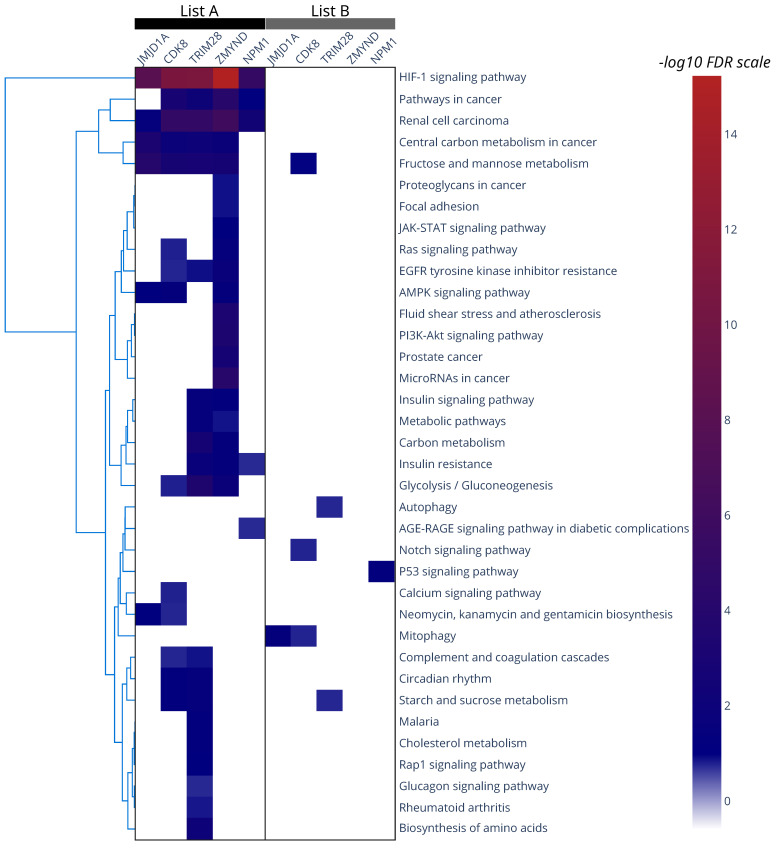
Heatmap of KEGG pathway analysis of common genes between JMJD1A, CDK8, TRIM28, ZMYND8 or NPM1 (as indicated) and LIST A (**left** panel) or LIST B (**right** panel). The -log10 FDR values for the various pathways were plotted for each of the overlapping gene sets. An FDR cut-off of 0.05 was used for statistical significance. Clustering was performed using the Nearest Point algorithm with Euclidean distance from SciPy [117]. The heatmaps were plotted using the python Plotly package (Plotly Technologies Inc. Collaborative data science, Montréal, QC, Canada, 2015. https://plot.ly accessed on 31 January 2023).

**Figure 3 cells-12-00798-f003:**
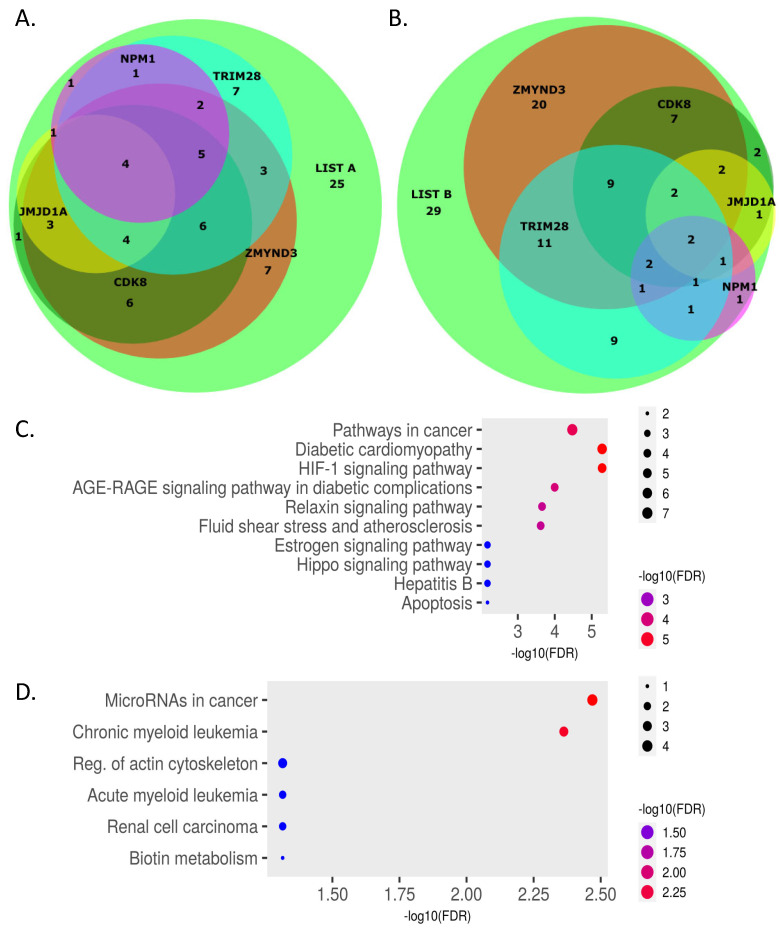
(**A**,**B**): Venn diagrams depicting the overlap between List A (**A**) or List B (**B**) genes and the five co-activator-dependent gene sets as indicated. (**C**,**D**) Dot plots of KEGG pathway analysis of List A (**C**) or List B (**D**) genes that do not overlap with any coregulator-dependent gene sets.

**Table 2 cells-12-00798-t002:** Comparative analysis of HIF-1-interacting co-activator-dependent gene target dataset with List A and List B of HIF-1-dependent gene targets.

HIF-1 Co-Activator	Ref.	Number of		
		Co-Activator- Dependent Genes	Common Genes with List A (83 Genes)	Common Genes with List B (109 Genes)
ZMYND8 #	[53]	603	45 *	62 *
CDK8	[54]	168	34 *	31 *
TRIM 28	[55]	1101	34 *	42 *
NPM1 #	[58]	436	19 *	12 *
JMJD1A #	[46]	224	15 *	13 *
TIP60	[44]	131	9 *	5
TET1 #	[49]	1044	16	10
Reptin	[42]	35	5	4
Pontin	[43]	66	2	2

* Denotes significant overlap between datasets; # Denotes HIF-1 direct targets.

## Data Availability

The data presented in this study is contained within the article and Appendix A.

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
