# Peer review of "Transcriptional Response to Hypoxia: The Role of HIF-1-Associated Co-Regulators"

_cells, 2023, doi:10.3390/cells12050798_

Round 1
Reviewer 1 Report
This is a timely and interesting review that provides insight, with comprehensive depth, into the role of HIF-1-associated co-regulators-
I have a few minor comments:
line 46: should it read “its translocation into the nucleus” instead of “its migration inside the nucleus”?
line 92-line 110. Is this all referring to references 14-19?
Figure1, figure legend: What is meant by “direct HIF-1 targets?” HIF-1 target genes?
Check that all abbreviations are spelled out one time.
For some HIF interactors, it is mentioned that there is direct binding of the proteins (e.g. Line 512 NPM1 and HIF). Should the reader therefore conclude that other bindings are indirect/in protein complexes? An additional column in Table 1 could be used to present information on the nature of the interaction for each co-regulator.
Font size in Figure 3 is rather small.
Author Response
Point 1: This is a timely and interesting review that provides insight, with comprehensive depth, into the role of HIF-1-associated co-regulators. I have a few minor comments: Our response: We thank the reviewer for appreciating our manuscript.
Point 2: line 46: should it read “its translocation into the nucleus” instead of “its migration inside the nucleus”? Our response: Yes, thank you, this is now corrected as suggested (line 48 of revised unmarked version)
Point 3: line 92-line 110. Is this all referring to references 14-19? Our response: Yes, this part is referring to references 14-19. To make this clearer, the relevant references were also added at the end of each sentence (lines 87-106 of revised unmarked version)
Point 4: Figure1, figure legend: What is meant by “direct HIF-1 targets?” HIF-1 target genes? Our response: Yes, and to make this clearer, the legend now reads “Genes directly regulated by HIF-1 …” (line 194 of revised unmarked version)
Point 5: Check that all abbreviations are spelled out one time. Our response: All abbreviations have been checked, missing ones have been added and all were spelled out only the first time they were mentioned.
Point 6: For some HIF interactors, it is mentioned that there is direct binding of the proteins (e.g. Line 512 NPM1 and HIF). Should the reader therefore conclude that other bindings are indirect/in protein complexes? An additional column in Table 1 could be used to present information on the nature of the interaction for each co-regulator. Our response: The reviewer is right to wonder. However, in almost all cases, the association between a co-regulator and HIF-1/2αwas demonstrated using co-immunoprecipitation experiments or pull-down assays using cell extracts, methods that do not formally prove direct binding since they cannot exclude the possibility of indirect association mediated by a third protein. In all these cases, we were careful to use expressions such as “association”, “interaction” or “complex formation”, despite of what might have been claimed in the original papers. Only in three cases, those of NPM1, FBP1 and PARP1, direct binding was formally demonstrated using purified recombinant or in vitro translated proteins or protein fragments. Therefore, the expression “direct binding” is only used in these cases and its implication is now clearly explained in the relevant parts of the text (lines 473-4, 519-21 and 533-4 of revised unmarked version). We would rather not add an additional column in Table because it is already too crowded and because we think that a reader interested to know the exact methodology, which was used to demonstrate an association, should consult the original papers.
Point 7: Font size in Figure 3 is rather small. Our response: Font size in Figure 3 was increased

Reviewer 2 Report
In this paper, the authors summarize the role of HIF-1 and its co-activators in the hypoxic response, and they argue that only a fraction of these genes are true direct targets of HIF-1, with the majority of them only indirectly regulated by HIF-1 through HIF-1-dependent induction of other transcriptional regulators.
While sections 1 and 2 provide good overall background, they do not cite separate references in each sentence. For example, in Section 2, the sentences " Second, the majority of hypoxia responsive genes did not contain a detectable HIF-binding site in their proximal promoter, although the majority of HIF-1-binding sites were localized in close proximity to genes." and "Third, less than 1% of the DNA promoter sequences containing the core RCGTG motif bound HIF-1 or HIF-2 and extended sequence preferences beyond the core motif could not explain the lower than-predicted number of observed HIF-1-bound sites, raising the issue of how productive HREs are selected." etc. need to be clarified as to which paper they are quoting. Please check throughout the paper.
Section 3 discusses the HIF-1 interacting coactivators one by one, but it is too long and redundant. Section 3 should be shortened to less than half its length. Also, each cofactor that appears in Section 3 would be better summarized in a table.
Section 4 is unique and excellent. However, I found it a bit annoying to read because of the back and forth between the supplementary figure and the main text, so it would be better if it could be improved.
Author Response
Point 1: In this paper, the authors summarize the role of HIF-1 and its co-activators in the hypoxic response, and they argue that only a fraction of these genes are true direct targets of HIF-1, with the majority of them only indirectly regulated by HIF-1 through HIF-1-dependent induction of other transcriptional regulators. Our response: We thank the reviewer for his constructive comments that follow.
Point 2: While sections 1 and 2 provide good overall background, they do not cite separate references in each sentence. For example, in Section 2, the sentences " Second, the majority of hypoxia responsive genes did not contain a detectable HIF-binding site in their proximal promoter, although the majority of HIF-1-binding sites were localized in close proximity to genes." and "Third, less than 1% of the DNA promoter sequences containing the core RCGTG motif bound HIF-1 or HIF-2 and extended sequence preferences beyond the core motif could not explain the lower than-predicted number of observed HIF-1-bound sites, raising the issue of how productive HREs are selected." etc. need to be clarified as to which paper they are quoting. Please check throughout the paper. Our response: We apologize for this omission. References have now been added after each relevant sentence in Section 2 and the rest of the paper.
Point 3: Section 3 discusses the HIF-1 interacting coactivators one by one, but it is too long and redundant. Section 3 should be shortened to less than half its length. Also, each cofactor that appears in Section 3 would be better summarized in a table. Our response: Section 2 is the main section of our review article as it gives both background information and the proposed mechanisms of action for the presented HIF-1-interacting coactivators. We think that this discussion is not redundant and it is necessary for the reader, who is not an expert on transcription, chromatin modification and remodeling, RNA Pol II regulation etc., in order to understand and appreciate the complexity of the hypoxic transcriptional response. Therefore, we believe that dramatically shortening this section will severely diminish the comprehensive depth of our review, which is praised by reviewer 1. Nevertheless, in order to follow the reviewer’s suggestion and make Section 2 more concise, we shortened it as much as possible by removing any information not directly relevant to the hypoxic response. We also removed the information on the cell types in which the HIF-1/co-activator interaction was studied and on HIF-1α domain involvement in each case. This information is now presented exclusively in Table 1, which is also referenced throughout the paper.
Point 4: Section 4 is unique and excellent. However, I found it a bit annoying to read because of the back and forth between the supplementary figure and the main text, so it would be better if it could be improved. Our response: To improve this issue, a part of Supplementary Figure 1 (Panels in F) has now been included in Fig. 3 as Fig. 3C & D. In this way, Supplementary Figure 1 is now only mentioned twice in the text and it only includes the KEGG pathway analysis results that are also presented in a more condensed form in Fig. 2.

Round 2
Reviewer 2 Report
The authors have addressed almost all of my concerns, and I have no further points.